# Design of a Self-Expanding Stent Mechanism Enacted by Fluid Pressure Difference

**Ming-Yen Chang, Hsing-Hui Huang *** and **Chia-Kai Lu**

Department of Vehicle Engineering, National PingTung University of Science and Technology,
Pingtung 91201, Taiwan; chang.mingyen@mail.npust.edu.tw (M.-Y.C.); m9938009@mail.npust.edu.tw (C.-K.L.)
* Correspondence: sanlyhuang@mail.npust.edu.tw

**Abstract:** In this study, the design of a metal stent which expands under the influence of a difference in hydraulic pressure is presented. Using the action of existing stents as a reference, the joint of the metal stent model is joined, to lock its own elastic force. The maximum energy storage formula was applied to determine if the joint could withstand the elastic force that is integral to the stent model. Simulations of the stent, under the influence of differences in hydraulic pressure, were performed. During simulation, the fluid pressure, the width of the joint of the stent, the angle of the pipe model, and some other parameters, were changed to determine their impact on the stent joint and to explore the differences.

**Keywords:** self-expanding mechanism; finite element analysis; hydraulic pressure

## 1. Introduction

Technological advances have resulted in the development of many self-expanding mechanisms, where expansion is triggered externally. An ordinary umbrella is such an example. When the button on the handle is pressed, a hidden spring exerts a force, which opens the umbrella. The emergency ejection seat in a fighter jet is another example of an expanding mechanism that is triggered by a push button. There are also many children's toys with similar features. An inflatable ball is expanded using external force. Cardiovascular stents are a typical example of expanding stents. These stents can be expanded by balloons or can be self-expanding. The use of balloon expanded stents is a mature technology. However, self-expanding stents usually require the removal of a binding device, after the stent has been positioned in the desired location in the lumen of the vessel, before the stent will expand under its own elastic force. The design of the stent needs to take the structure and the type of material used, its processing, surface treatment, and other details into careful consideration. In addition to having the strength needed to dilate and support the vessel, other factors, such as a change in length during expansion, also need to be predictable and controlled.

Very little has been published on stent expansion that relies on liquid pressure. This review will, therefore, be done in two parts: the mechanical model, followed by a simulation analysis.

Both Nuttall [1] and Pollard [2] separately studied the ejection seat systems in fighter jets in 1971. These systems have expandable mechanisms that are activated by a button. In 1983, Cragg et al. [3] made a stent with coiled nitinol wire, which has a heat-sensitive memory. These coils were easy to manufacture and had good elasticity. However, they were not very strong. In 1985, Wright et al. [4] described the first self-expanding, spring-loaded stent. In 1987, Rousseau et al. [5] described a grid self-expanding stents made of stainless steel. Withdrawal of a protective cover, once the stent was positioned in the vessel, allowed it to expand. Chua et al. [6,7] analyzed the interaction between the balloon and the slotted tube stent. In their studies, half and quarter models are used in the finite element analysis. Etave et al. [8] studied the mechanical properties of both tubular and coil stents by ABAQUS. Dumoulin and Cochelin [9] studied the behavior between radial and

longitudinal recoils, then minimized the weakness of the structure of balloon-expandable stents. Patent [10] described a stent design in which two compensating structures combined in a way that resulted in no change in length upon expansion. In 2006, Garcia et al. [11] described a self-expanding stent that could treat general malignant obstruction caused by colon cancer, which was held in the compressed state by a cord. When the cord was pulled and removed, the stent expanded. In 2007, Lee et al. [12] described a self-expanding stent for the treatment of a gastric outlet obstruction caused by gastric cancer. The binding wire was withdrawn to allow the stent to expand after it had been introduced into the affected area using a guide wire. In the same year, Patel and Ananthasuresh [13] analyzed the expandable mechanical sphere. This is a toy that can be expanded by means of an external force. The model returns to its original state when the force is removed. In 2011, Wright [14] also designed a stent with an external cover or covering thread.

With regards to the literature on the introduction of the model to CAE software's research of the expanding motion, by means of the finite element method, most analyses were done by simplification or by taking aliquots of the model. In 2000, Dumoulin and Cochelin [15] simplified the Palmaz stent into a unit model and explored the differences in the stress and strain in 2D and 3D models. They also streamlined stent simulation and shortened computing time in order to obtain results that showed variance consistently smaller than 5%. In other words, the stent could be looked at as a plane to facilitate design and analysis. The Palmaz stent created by Dai [16] in 2004 was known for its symmetry. The model was divided into six parts to reduce the analytical and computing time needed. In addition, it was noted that the back pressure effect of the lining occurred while the stent was expanding, and it would have an effect on the extent of the deformation and retraction of the stent. In 2008, Ni et al. [17] used an artificial neural network to make an intelligent forecast of the deformation of expanding stents. This approach was highly adaptive and could be used to optimize the design of the stent structure. This was a new approach to the investigation of deformation in expanding stents. In 2009, Li et al. [18] used an optimization theory instead of a conventional approach and carried out a finite element analysis of a stent. Study results showed that their optimized stent design had a significantly better mechanical performance. In 2011, Malvè et al. [19] used a finite element approach to simulated analysis. Their results showed that the presence of a stent helps to prevent the muscular deformation of an airway and that the analysis of stress distribution can be used to determine the location for placement and the orientation of a stent.

Clearly, not much work has been done on the application of an expanding stent model. Expanding stents discussed in the literature were mainly of a type that needed to be artificially expanded. In this study, the design of a stent model which can be expanded using hydraulic pressure is offered. The expanding ring design that is introduced here, takes advantage of a hydraulic pressure difference; additionally, the material used was metal, which can be applied to smooth vessels or to self-expanding vascular stents.

## 2. Basic Principle

To address the issue of lengthy computing times for analysis, explicit integration was used to simulate the expansion and deformation in a fluid pressure self-expanding mechanism. Direct integration is often used to solve the equation of motion as part of the second-order ordinary differential equation; however, to solve the equation of motion, direct integration is applied, regardless of the change in pattern. The direct integration method can also be explicit or implicit. With the finite element approach, explicit integration does not involve convergence. There is no need to determine whether convergence is present or not in the analysis or calculation of the next time increment. However, when a time increment in an analysis is relatively small, iterations need to occur more frequently. On the other hand, implicit integration involves convergence, such as plastic deformation and contact. Under excessive contact, each time increment needs to be calculated many times until contact convergence occurs.

With the finite element approach, the governing equation is required to obtain the shift vector of each node a(t) and then the required stress and strain. For a dynamic system, the equation of motion is a second-order ordinary differential equation [20] as in (1) below:

$$M\ddot{a}(t) + c\dot{a}(t) + Ka(t) = Q \tag{1}$$

In Equation (1), M is the system's mass matrix; $\ddot{a}(t)$ is the acceleration vector of the system node; C is the system's damping matrix; $\dot{a}(t)$ is the speed vector of the system node; K is the system's rigidity matrix; a(t) is the shift vector of the system node; Q is the load matrix of the system.

The shift solution to time, in explicit integration, satisfies the equation of the motion of time (t) as follows:

$$M\ddot{a}_t + c\dot{a}_t + Ka_t = Q_t \tag{2}$$

Speed and acceleration are indicated as (3) and (4), and are introduced into (2) to obtain (5), as follows:

$$\dot{a}_t = \frac{1}{2\Delta t}(-a_{t-\Delta t} + a_{t+\Delta t}) \tag{3}$$

$$\ddot{a}_t = \frac{1}{\Delta t^2}(a_{t-\Delta t} - 2a_t + a_{t+\Delta t}) \tag{4}$$

$$\left(\frac{1}{\Delta t^2}M + \frac{1}{2\Delta t}C\right)a_{t+\Delta t} = Q_t - \left(K - \frac{2}{\Delta t^2}M\right)a_t - \left(\frac{1}{\Delta t^1}M - \frac{1}{2\Delta t}C\right)a_{t-\Delta t} \tag{5}$$

$a_{t+\Delta t}$ is obtained from (5) and it can be seen that the solution $a_{t+\Delta t}$ is based on the motion status of time (t). In this study, the explicit integration method is used to simulate the expansion and deformation of the mechanism in fluid pressure to solve the problem of the long calculation times in the analysis.

## 3. Conceptual Design

The design process used in this study is shown in Figure 1. Prior to conceptual design, the design demands and the criteria of the model have to be set. The basic stress analysis of the model should help obtain related information. In Figure 2, which is a simulated environmental cross section, yellow, blue, and brown represent the piping, the liquid, and the stent, respectively. An understanding of the possible force applied to the stent as it goes through the pipe, and the distribution of stress withstood by the model as the stent passes through the narrow part, are used as a basis for determining the ring design.

The materials of the pipe are set to be either elastic or rigid. When the pipe model is elastic, simulation shows that, when the stent enters the narrow part (see Figure 3a), a shrinking flow field and a greater pressure cause stress at the front end of the model, the pipe lining is squeezed by the liquid pressure, and the pipe swells. As the stent continues to move forwards, entering the narrow part, the pipe bounces back to squeeze the rear end of the stent and excessive stress occurs, as shown in Figure 3b. The same simulation, repeated with the material of the pipe set to rigid, shows that as the stent enters the narrow part (as shown in Figure 4)—due to a shrinking flow field and greater pressure—the front end of the stent undergoes greater stress. As the stent continues forwards and enters the narrow part, the stress does not show much variation. After the ring passes through the narrow part of the pipe, regardless of the material, stress occurs in the narrow part and stress distribution appears to be ring-shaped. This focus of the stress on the ring as it passes through the narrow part of the pipe, the design of the fastener or latch, and the changes in the local cross-section area, are all taken into account in the simple solution shown in Figure 5.

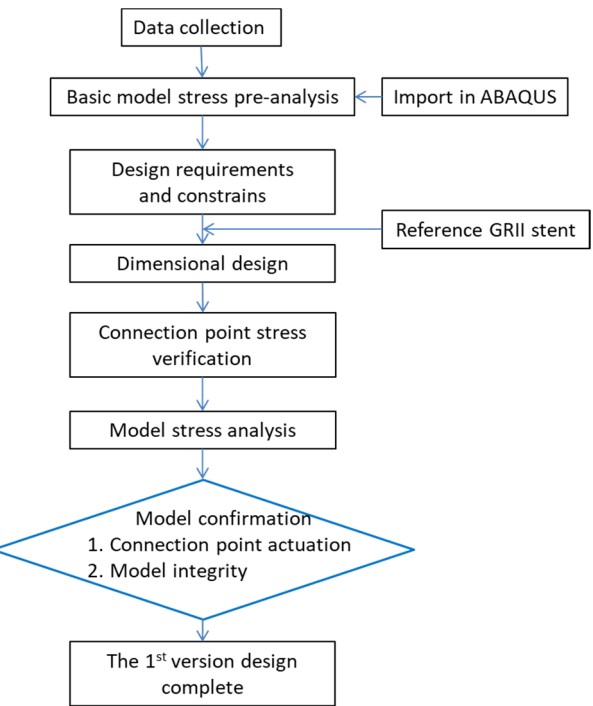

**Figure 1.** Stent design procedure.

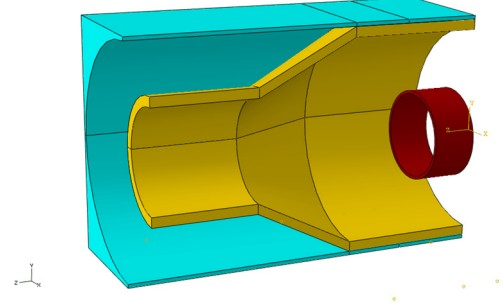

**Figure 2.** Simulated environmental cross-section.

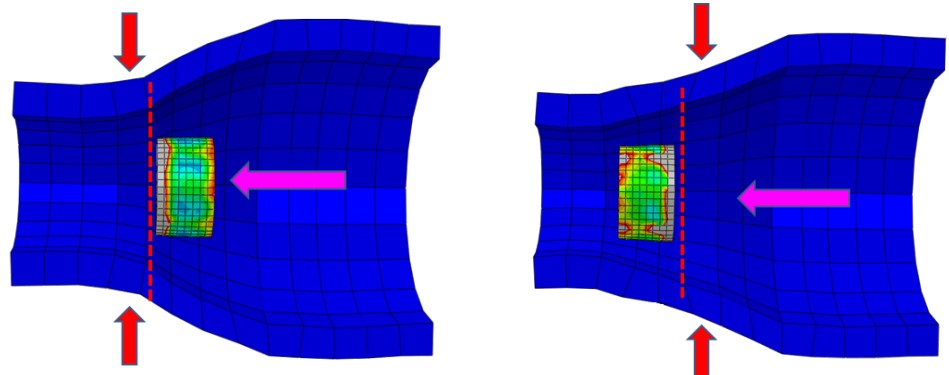

(**a**) Before entering the narrow part of the pipe     (**b**) After entering the narrow part of the pipe

**Figure 3.** Elastic simulated environment.

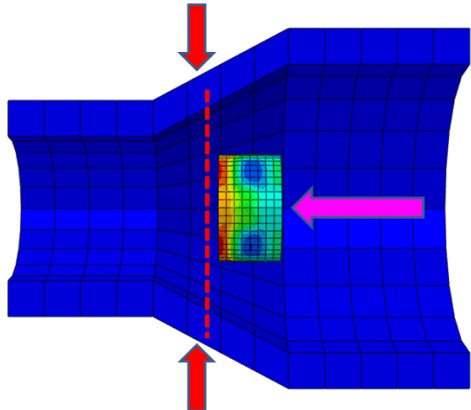

**Figure 4.** Rigid simulated environment.

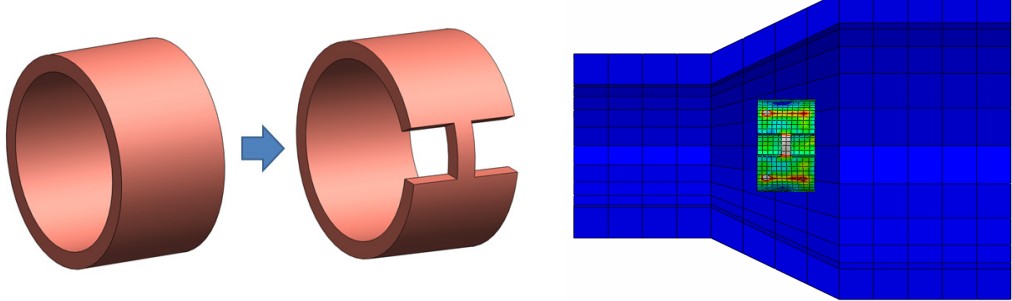

**Figure 5.** Changes in local cross-section area (**left**) and stress distribution on the stent (**right**).

### 3.1. Design Requirements and Constraints

The results of a basic stress analysis of the stent can be summarized as follows:

1. The stent must withstand different fluid pressures, exerted in different places on the model.
2. Fasteners or latches are needed in order to lock the intrinsic elastic force of the stent.
3. Stress positions on the stent can be controlled by changes the cross-section area.

Design constrains may be summarized as follows:

1. The stent must retain its circular shape after expansion to maintain smooth flow through the vessel.
2. The stent should have an expansion ratio of 2–3 times.
3. Variation in axial length during or after expansion should be avoided.

Model 1 was derived from taking the GR-II$^{TM}$ stent [21] as a starting point, following consideration of the design requirements and the constrains of motion of a stent. From a side view, the opening and retraction of the model is activated in a plane vortex manner. Model 1 (as shown in Figure 6a) is compressed and has been designed with a fastener or latch mechanism to lock and prevent expansion, as a result of intrinsic elastic force. The joint is shown in Figure 6b. A further simplification gives Model 2, as shown in Figure 7.

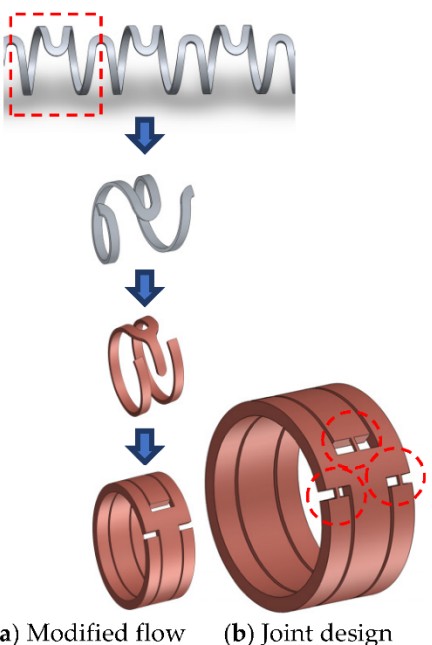

(**a**) Modified flow     (**b**) Joint design

**Figure 6.** Stent design Model 1.

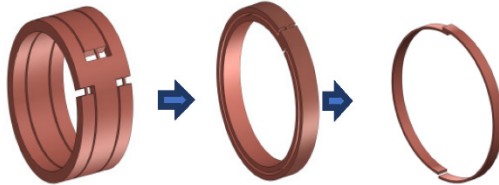

**Figure 7.** Model 2 illustration.

### 3.2. Creation of a Geometric Model

The metal stent model and the pipe model that were created for this study have small and large diameters. This is necessary for the generation of a difference in pressure in a fluid. According to Pelton [22] this is around 2~3 mm and 6~9 mm before and after the expansion of a vascular stent. The metal stent model proposed here is held in an initial retracted state by a fastener, which restricts its elastic force. The model before expansion had a diameter of 3.2 mm and the ring was 0.25 mm thick. After expansion, the diameter was 6.5 mm and it was 0.1 mm thick. The width was maintained at 0.5 mm before and after expansion. Figures 8 and 9 show the geometric details of the metal stent and those of the joint. The pipe model was 5.75 mm long. The inlet was 8 mm, with reference to the size of a coronary artery [22], and the outlet was 4 mm. Figure 10 shows the geometric cross-section of the pipeline. Next, a liquid model 14 mm long, 14 mm wide, and 6 mm high was derived by the application of Eulerian theory with ABAQUS. Then, the metal stent and pipe model were immersed in liquid, see Figure 11.

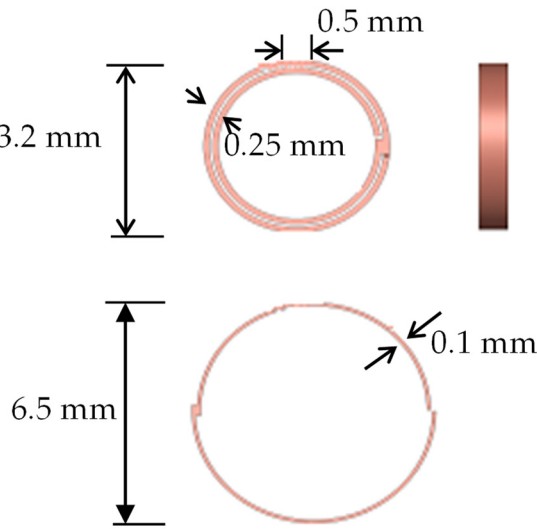

**Figure 8.** Geometric dimensions of the metal stent model.

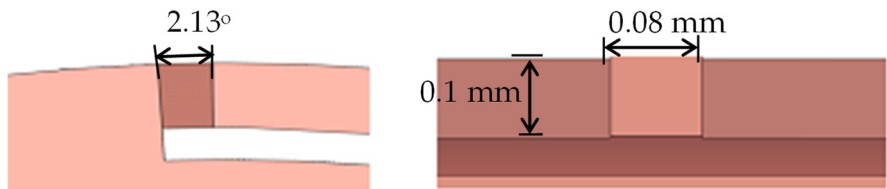

**Figure 9.** Geometric dimensions of the metal stent joint.

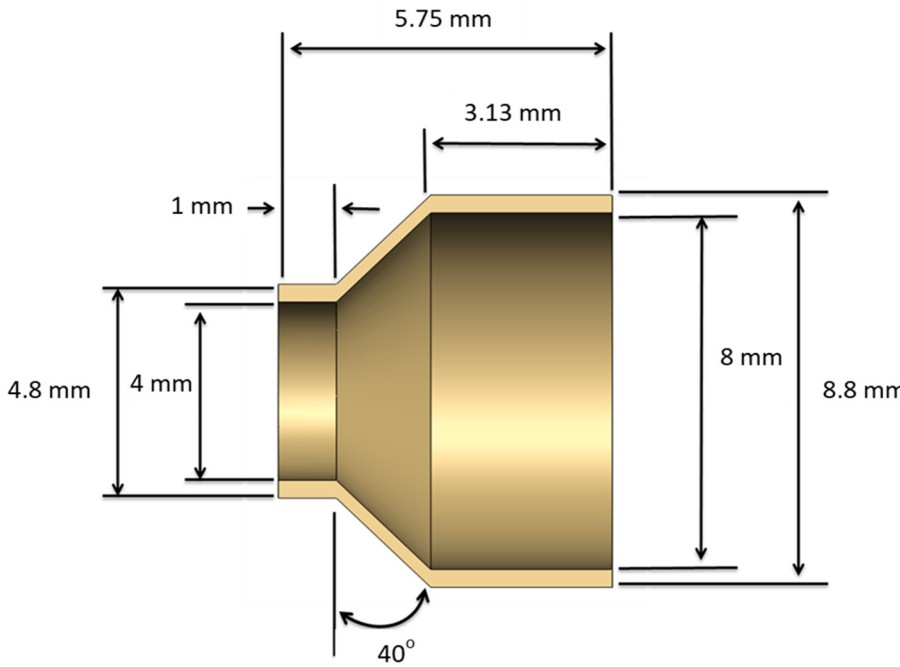

**Figure 10.** Geometric dimensions of the pipe model (cross-section).

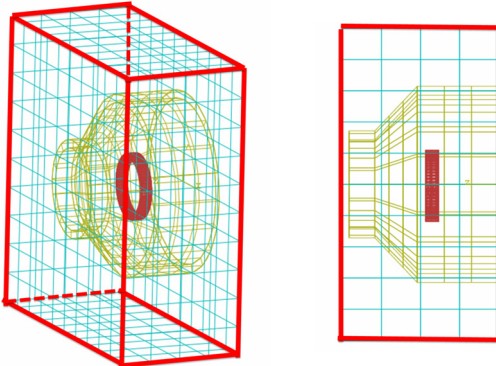

**Figure 11.** Liquid model illustration.

### 3.3. Material Properties

Aluminum alloy 6061-T6 was selected for the metal stent model. The mechanical parameters of the material were obtained by a tensile experiment. Post-converted stress and strain data were used to produce the stress–strain curve that is shown in Figure 12. Table 1 shows the material parameters of aluminum alloy 6061-T6. Table 2 shows plasticity parameters of aluminum alloy 6061-T6. When fluid flows, a pressure difference occurs, due to the change in diameter. For this study, the pipe model was set to be that of a PVC water pipe [23]. The material parameters are shown in Table 3. The liquid, on the other hand, was set using the parameters provided in the ABAQUS document. Table 4 shows the liquid parameters. All the material models are assumed to be homogeneous and isotropic.

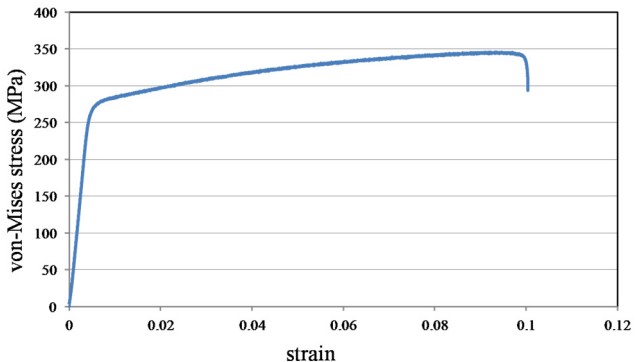

**Figure 12.** Stress–strain curve for aluminum alloy 6061-T6.

**Table 1.** Material parameters of aluminum alloy 6061-T6.

|  | Aluminum Alloy 6061-T6 |
| --- | --- |
| Elasticity coefficient | 68,900 (MPa) |
| Poisson's ratio | 0.33 |
| Material density | 2700 (kg/m$^3$) |

**Table 2.** Plasticity parameters of aluminum alloy 6061-T6.

| Yield Stress (MPa) | Plastic Strain |
| --- | --- |
| 277 | 0 |
| 295 | 0.012 |
| 315 | 0.03 |
| 335 | 0.059 |
| 345 | 0.088 |

**Table 3.** Material parameters of pipe model.

|  | PVC |
| --- | --- |
| Elasticity coefficient | 3400 MPa |
| Poisson's ratio | 0.33 |
| Material density | 1380 (kg/m$^3$) |

**Table 4.** Liquid parameters.

|  | Liquid |
| --- | --- |
| Mass density | 996 (kg/m$^3$) |
| Viscosity coefficient | $1 \times 10^{-6}$ (kg·m$^{-1}$·s) |

### 3.4. Boundary Condition for the Simulation

In this study the metal stent was located in the large diameter area of the pipe and the whole model was immersed in fluid to simulate activation of the metal stent when covered in fluid. Meanwhile, contact is set between all models. In order to prevent forward shifting of the pipe as a result of squeezing from the fluid due to pressure difference, fixation at the inlet of the pipe model is done in all directions. The metal stent model used in this simulation was the size of a vascular stent and the fluid pressure was that of hypertensive human blood pressure at 0.024 MPa [23]. Figure 13 is the illustration of the liquid load applied during the finite element analysis of runner pressure difference of the metal stent. The model used was dynamic and the simulation and analysis were carried out over 1.5 ms, which is the time needed for the metal stent to pass through the narrow part of the pipe.

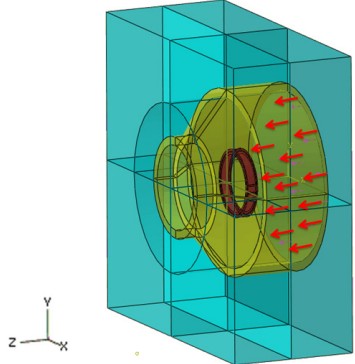

**Figure 13.** Runner pressure load illustration.

### 3.5. Joint Model Verification

The initial state of the metal stent model is compressed. A joint is needed to lock the model to prevent expansion from the intrinsic elastic force. If the joint in the metal stent cannot withstand this intrinsic force, it will expand before being subject to the liquid pressure required for activation. To ensure this requirement, it is necessary to test the joint to ensure it can withstand the intrinsic elastic force in the metal stent.

From Les [23] the maximum energy of a flat coiled spring can be obtained using the following formula:

$$\varepsilon = \frac{WS^2 tl}{6E} = \frac{VS^2}{6E}$$

*W*: Mainspring width
*S*: Maximum permissible stress
*t*: Mainspring thickness
*l*: Mainspring length
*V*: Mainspring volume

*E*: Young's modulus

*ε*: Maximum energy of mainspring

The total volume (*V*) of the metal stent model in this study is 0.9291 mm$^3$ and the maximum permissible stress (*S*) is 345 MPa. Young's modulus (*E*) is 68900 MPa. Introducing the above parameters into the formula below:

$$\varepsilon = \frac{VS^2}{6E} = \frac{0.9291 \text{ mm}^3 * \left(345 \frac{\text{N}}{\text{mm}^2}\right)^2}{6 * 68900 \frac{\text{N}}{\text{mm}^2}} = 0.26750 \text{ N.mm}$$

Calculation showed the value of the maximum energy (*ε*) to be 0.26750 N.mm. ABAQUS was then used to analyze the load and determine if the joint could withstand a torque of 0.26750 N.mm without causing damage. Figure 14a shows the geometric dimensions of the metal stent joint model. The joint is made of the same metal as the ring (aluminum alloy 6061-T6). An eight-node hexagonal entity element grid was selected for the joint model 3D depiction. One side of the model was fixed and a torque of 0.26750 N.mm was applied to the other side in the y direction. Analysis showed the maximum stress generated to be 277.8 MPa. Figure 14b shows the stress distribution over the joint model, and Figure 15 is a stress and time curve at the metal stent joint of 0.08 mm × 0.1 mm. Because the generated maximum stress is slightly more than the yield strength of 277 MPa, slight deformation might occur at the joint, but this will not exceed the tensile strength of the material at 345 MPa. This demonstrates that, when the intrinsic elastic force of the metal stent model is constrained by the joint, the joint will not be damaged as a result of excessive elastic force in the metal stent.

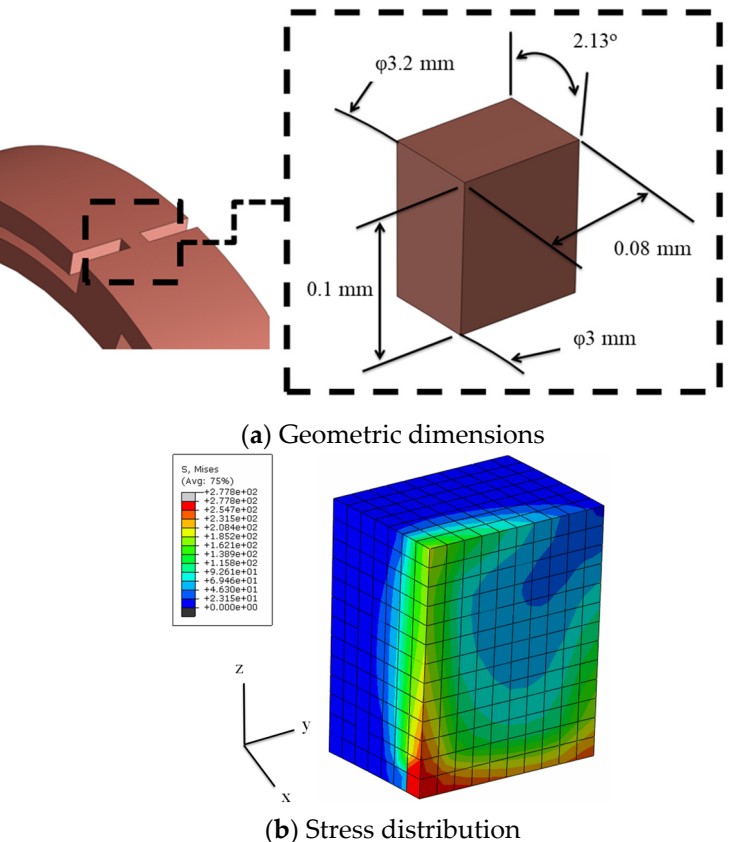

(**a**) Geometric dimensions

(**b**) Stress distribution

**Figure 14.** Geometric dimensions and stress distribution over the metal stent joint.

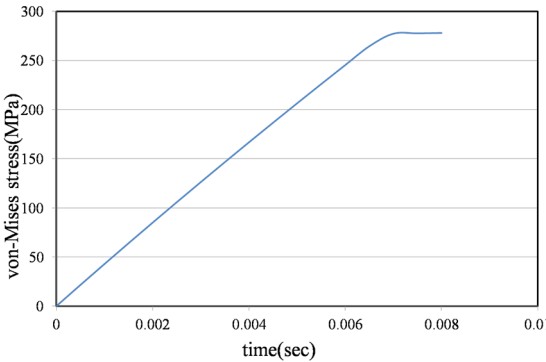

**Figure 15.** Stress and time curve at a 0.08 mm × 0.1 mm joint.

## 4. Analysis Results

The impacts of all the features were compared using different parameters, such as liquid pressure, metal stent joint width, and pipe model angle.

### 4.1. Metal Stent Stress Analysis

A metal stent model with a joint of 0.08 mm × 0.1 mm was placed in a pipe model with a pipe angle of 40°. Simulation showed that, when the joint of the metal stent model was at 0.645 ms, the stress had already reached the tensile strength of 345 MPa. Therefore, a stress–time simulation was explored where the grid elements had reached the tensile strength first. Figure 16b shows the stress distribution when the metal stent joint was at 0.645 ms. It clearly shows that the stress was focused in the joint, the metal stent itself not being subject to much stress. Figure 16a shows that the metal stent joint breaks when it is in the narrow part of the pipe model. This simulation showed the stress–time relationship when the grid elements of the joint had reached the tensile strength first. Figure 17 shows a stress–time curve of the metal stent joint model. From this curve, it can be seen that, when the metal stent starts to push forwards, the joint is gradually subjected to stress generated by the moving liquid. By 0.645 ms, the metal stent joint had already reached the tensile strength and could not withstand any additional stress and breaks.

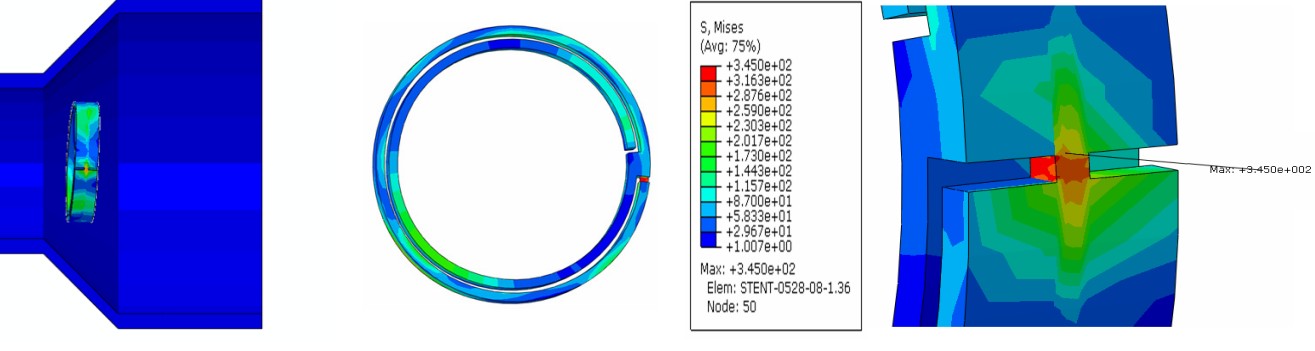

(**a**) Position of joint upon breakage          (**b**) Stress distribution

**Figure 16.** Local enlarged stress distribution of metal stent joint 0.08 mm × 0.1 mm model.

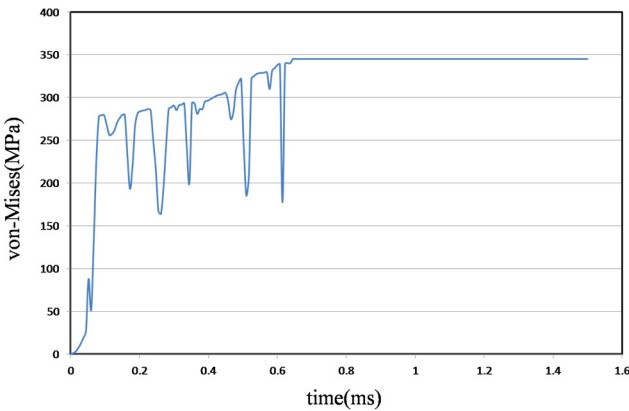

**Figure 17.** Stress–time curve of metal stent joint 0.08 mm × 0.1 mm model.

To further investigate joint breakage, ductile damage was added as a material parameter to the metal stent model, and simulation analysis was repeated. Figure 18 shows the metal stent joint breakage plot, which clearly shows breakage of the joint at 0.645 ms, which occurred as a result of stress focused at the joint.

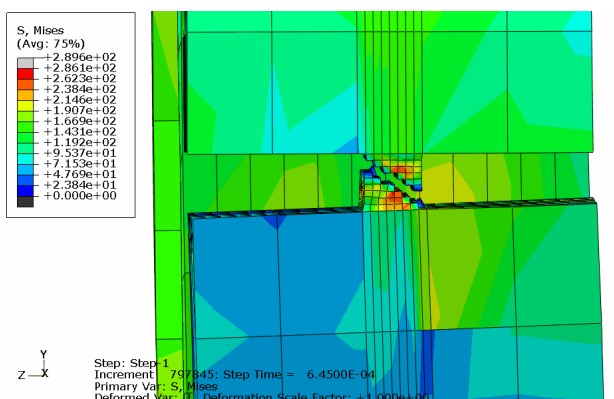

**Figure 18.** Metal stent joint breakage plot.

*4.2. Comparison of Different Liquid Pressures*

The amount of stress that the metal stent joint was subjected to was investigated using different liquid pressures, and a metal stent joint of 0.08 mm × 0.1 mm was used in the analytical model. Two liquid pressures were used in the simulations and analysis—normal blood pressure at 0.16 MPa and hypertensive blood pressure at 0.24 MPa. In addition, the maximum stress position of the joint was plotted as a stress–time curve, as shown in Figure 19. It can be clearly seen that at a liquid pressure of 0.024 MPa, the joint stress had already reached the tensile strength at 0.645 ms; moreover, at a liquid pressure of 0.016 MPa, the joint stress reached the tensile strength at 0.705 ms. Comparison between the two clearly shows that the force on the metal stent joint that was simulated by normal blood pressure had a breakage time later than that with hypertensive blood pressure. Normal blood pressure is associated with a breakage time 4% lower than that with hypertensive blood pressure.

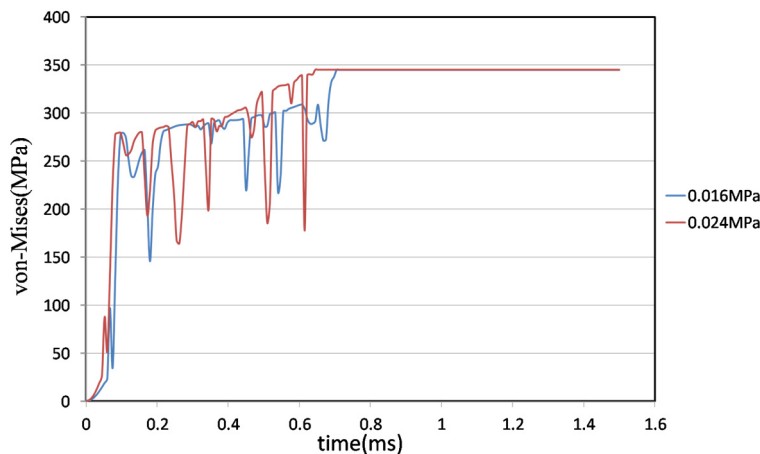

**Figure 19.** Stress–time curve of metal stent joint at different liquid pressures.

### 4.3. Comparison of Different Metal Joint Widths

Liquid pressure simulation analysis was carried out using three different metal joint widths (see Figure 20). The joint dimensions used were 0.06 mm × 0.1 mm, 0.08 mm × 0.1 mm, and 0.1 mm × 0.1 mm. Before analysis, it was necessary to determine whether each joint could withstand the intrinsic elastic force of the metal stent. The total volume ($V$) of the metal joints of the 0.06 mm × 0.1 mm model and the 0.1 mm × 0.1 mm model were 0.9289 mm$^3$ and 0.9292 mm$^3$, respectively. $V$ was introduced into the maximum storage energy formula, and it was found that the maximum storage energy ($\varepsilon$) of the 0.06 mm × 0.1 mm model joint was 0.26744 N.mm, and that of the 0.1 mm × 0.1 mm model was 0.26753 N.mm. Joint load simulation analysis was carried out and it was found that the maximum stress generated by the 0.06 mm × 0.1 mm model was 278.48 MPa, and that of the 0.1 mm × 0.1 mm model was 277.57 MPa. Neither exceeded 345 MPa, which was the tensile strength of the material. In other words, the intrinsic elastic force of the two metal stent models would not damage the joint.

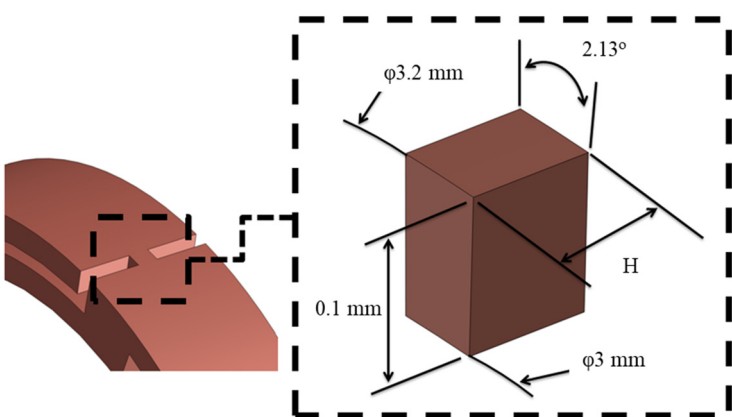

**Figure 20.** Width (H) variable illustration of the metal stent joint model.

After it had been found that the intrinsic elastic force of the metal stent could be maintained by the joint, models with the three different joint widths were introduced into ABAQUS and simulation analysis was done. The results showed that at 0.630 ms stress in the 0.06 mm × 0.1 mm model had already reached the tensile strength: Figure 21a shows the local enlarged stress distribution. When the 0.1 mm × 0.1 mm joint was at 0.683 ms, it had already reached the tensile strength: Figure 21b shows the local enlarged stress distribution. Figure 22 is a stress–time chart that shows a comparison of the three different joint widths. The breakage–time point was earlier in the narrow joints, as would

be expected. The 0.06 mm × 0.1 mm joint broke 1% earlier than the 0.08 mm × 0.1 mm joint, and the 0.1 mm × 0.1 mm joint broke 2.53% later than the 0.08 mm × 0.1 mm joint.

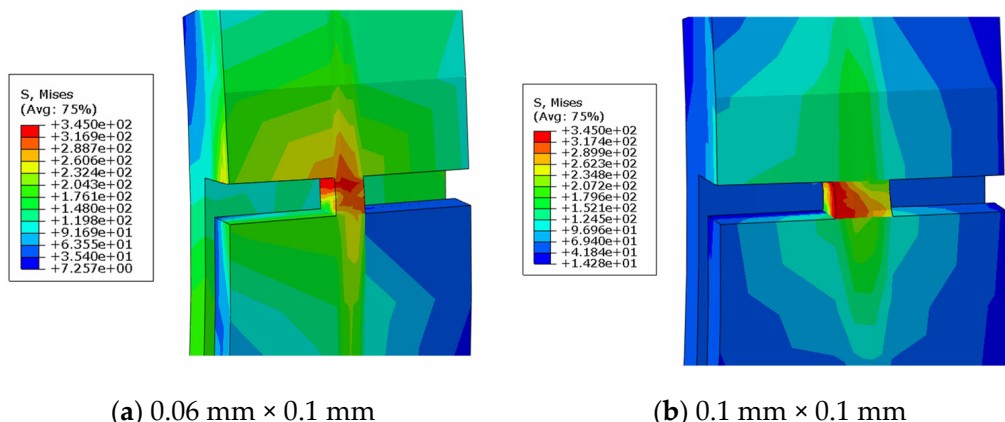

(**a**) 0.06 mm × 0.1 mm                    (**b**) 0.1 mm × 0.1 mm

**Figure 21.** Local enlarged stress distribution of metal stent joint model.

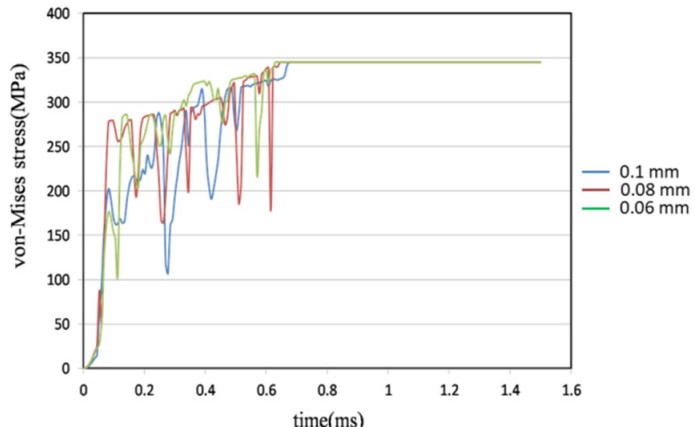

**Figure 22.** Stress–time curves for metal stent joints of different widths.

### 4.4. Comparison of Different Pipe Angles

Liquid pressure simulation analyses were carried out using three different pipe angles (G) (20°, 40°, and 60°), as well as with the three different joint widths, as before. Figure 23 shows the pipe model angle variable. Figure 24a shows the stress–time curve of three types of metal stents with a pipe angle of 20°. It can be seen that the 0.06 mm × 0.1 mm joint broke 6% earlier (at 9.323 ms) than the 0.08 mm × 0.1 mm joint at 0.413 ms, and the 0.1 mm × 0.1 mm joint broke 3.46% later (at 0.465 ms) than the 0.08 mm × 0.1 mm joint. Figure 24b shows stress–time curves of the three types of metal stents at a pipe angle of 60°. The 0.06 mm × 0.1 mm joint broke at 0.810 ms, which was 34.53% faster than the 0.08 mm × 0.1 mm joint at 1.328 ms; the 0.1 mm × 0.1 mm joint broke at 1.463 ms, which was 9% slower than the 0.08 mm × 0.1 mm model (which broke at 1.328 ms). It is clear that the joints will break sooner at smaller angles than at larger ones. Figures 25 and 26 show the stress distribution upon breakage for the three types of joints at pipe angles of 20° and 60°.

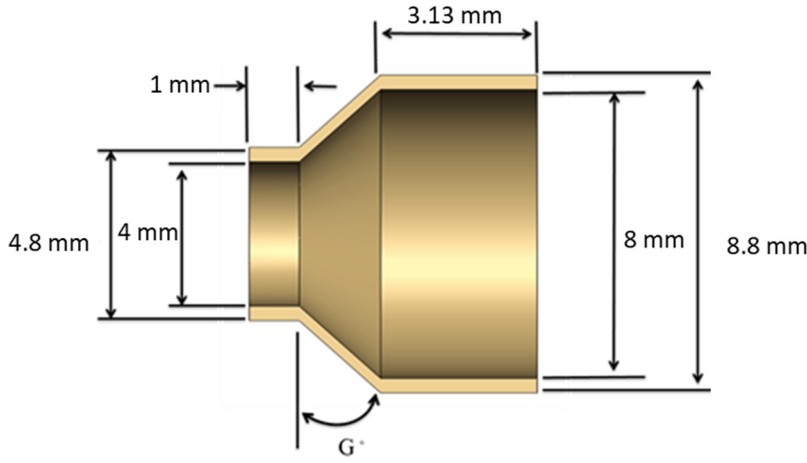

**Figure 23.** Pipe model angle variable G.

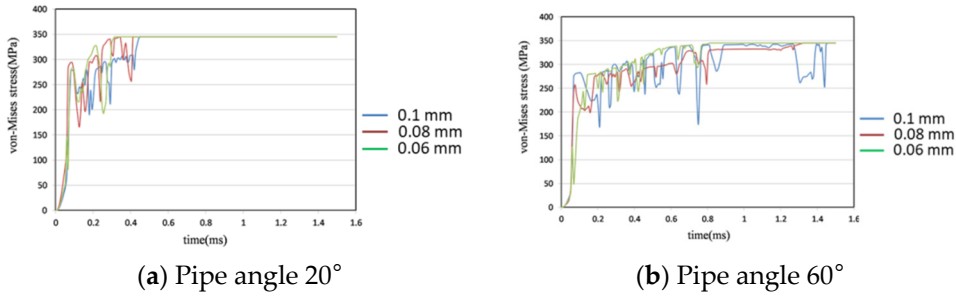

(**a**) Pipe angle 20°        (**b**) Pipe angle 60°

**Figure 24.** Stress–time curves for the 20° and the 60° pipe angles, with three sizes of joint.

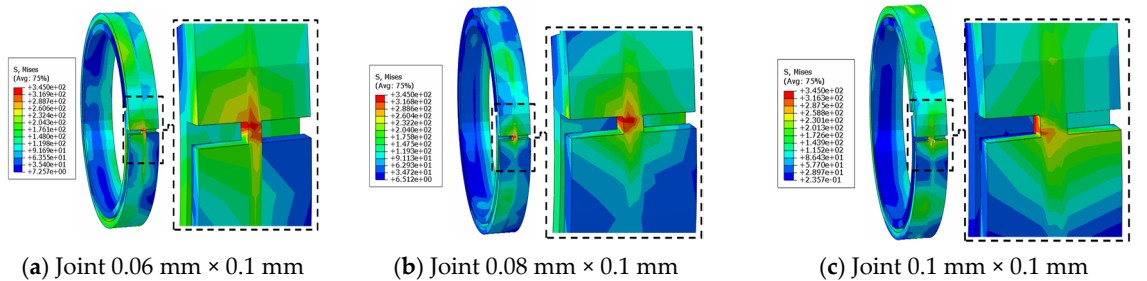

(**a**) Joint 0.06 mm × 0.1 mm     (**b**) Joint 0.08 mm × 0.1 mm     (**c**) Joint 0.1 mm × 0.1 mm

**Figure 25.** Stress distribution—20° pipe model.

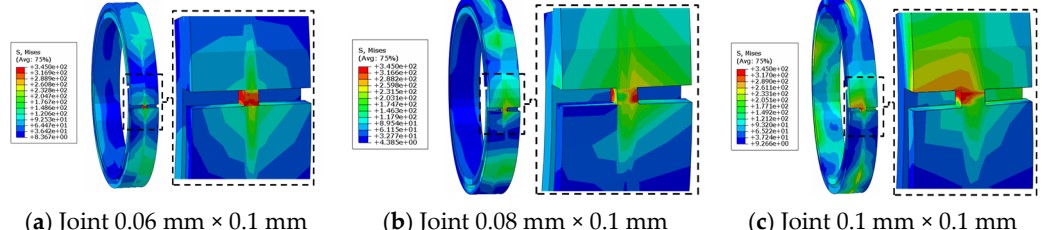

(**a**) Joint 0.06 mm × 0.1 mm     (**b**) Joint 0.08 mm × 0.1 mm     (**c**) Joint 0.1 mm × 0.1 mm

**Figure 26.** Stress distribution—60° pipe model.

## 5. Conclusions and Suggestions

In this conceptual design, the product is a metal stent in which intrinsic elastic force is contained by a joint that can be opened by fluid pressure. The maximum storage energy formula was used to verify whether the joint could withstand the intrinsic elastic force in the metal stent. Simulations were performed to verify various aspects of the design,

including liquid pressure simulation analysis. The stress analysis results of this study allowed the following conclusions to be made:

1.  Two different pressures were investigated—0.016 MPa and 0.024 MPa—using the same pipe model and a metal stent with a 0.08 mm × 0.1 mm joint. Under the lower pressure, the joint broke 4% later than at the higher pressure.
2.  Three different joint widths—0.06 mm, 0.08 mm, and 0.1 mm—were simulated and compared at a pipe angle of 40°. The 0.06 mm joint broke 1% sooner than the 0.08 mm joint and the 0.1 mm model broke 2.53% later than the 0.08 mm model. The breakage time of the metal stent joint is dependent on width.
3.  Three different pipe angles—20°, 40°, and 60°—with three different joint widths—0.06 mm, 0.08 mm, and 0.1 mm—were compared. At 20°, the 0.06 mm joint broke 6% earlier than the 0.08 mm joint and the 0.1 mm joint broke 3.46% later than the 0.08 mm joint. At 60°, the 0.06 mm joint broke 34.53% faster than the 0.08 mm joint and the 0.1 mm broke 9% more slowly than the 0.08 mm joint. The metal stent joints break more slowly with a shallower pipe angle.

In this study, only metal stent joint model breakage stress and parameter change analyses were done and the model has not been optimized. An optimized analysis of the model may be performed on the metal stent and post-expansion side angle; additionally, the sharp nature of the model may be modified with a simulation of the post-expansion status of the metal stent model.

**Author Contributions:** M.-Y.C., H.-H.H. and C.-K.L., conceived of the presented idea. We developed the theory and C.-K.L., performed the CAE. M.-Y.C. and H.-H.H. verified the analytical methods and supervised the findings of this work. All authors discussed the results and contributed to the final manuscript. All authors have read and agreed to the published version of the manuscript.

**Funding:** This research received no external funding.

**Institutional Review Board Statement:** Not applicable.

**Informed Consent Statement:** Not applicable.

**Conflicts of Interest:** The authors declare no conflict of interest.

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
