# Peer review of "Design of a Self-Expanding Stent Mechanism Enacted by Fluid Pressure Difference"

_applsci, doi:10.3390/app112110114_

Round 1

Reviewer 1 Report

The issue of stent implantation and all the dangers associated with it, although it is nothing new, is still an interesting topic for theoretical and practical considerations. The article is interesting and accessible to the reader, and describes the issue of the influence of blood pressure and the change in the cross-section of blood vessels on the opening of the stent. Considerations based on a simple model of a pipe with a variable cross-section and inclination angle of the walls illustrate well how complicated this crease is. I think the article is a good introduction to further theoretical and experimental considerations on more complex models.
My only critical remarks are that for better readability of the article, it would be good to present all the data in one system of units (SI).

For better readability of the article, it would be good to present all numerical data in the text, tables and figures in one system of units (SI - i.e. stiffness, viscosity, etc.)

Author Response

Thank you for your valuable suggestions.

The units of density and viscosity in the table have been modified to SI system.

Reviewer 2 Report

The reviewer thinks that - biocompatibility of the used material should be studied - comparison with stents made from shape memory alloys would be of interest - the maximum pressure/force induced by the stent after expansion should be limited.

Author Response

Thank you for your valuable suggestions. If the research continues, we may be able to use biocompatibility material as a more practical and ideal material.